# Promotion of Deoxycholic Acid Effect on Colonic Cancer Cell Lines In Vitro by Altering the Mucosal Microbiota

**DOI:** 10.3390/microorganisms10122486

**Published:** 2022-12-15

**Authors:** Yanpeng Ma, Yi Zhang, Ruize Qu, Xin Zhou, Lulu Sun, Kai Wang, Changtao Jiang, Zhipeng Zhang, Wei Fu

**Affiliations:** 1Department of General Surgery, Peking University Third Hospital, Beijing 100191, China; 2Cancer Center, Peking University Third Hospital, Beijing 100191, China; 3Laboratory of Metabolism, Center for Cancer Research, National Cancer Institute, Bethesda, MD 20892, USA; 4Key Laboratory of Molecular Cardiovascular Science, Ministry of Education, Department of Physiology and Pathophysiology, School of Basic Medical Sciences, Peking University, Beijing 100871, China; 5Center of Basic Medical Research, Institute of Medical Innovation and Research, Peking University Third Hospital, Beijing 100871, China; 6Center for Obesity and Metabolic Disease Research, School of Basic Medical Sciences, Peking University, Beijing 100871, China

**Keywords:** colorectal cancer, gut microbiota, bile acid, deoxycholic acid

## Abstract

Colorectal cancer (CRC) is the third most prevalent neoplasm and the second leading cause of cancer death worldwide. Microbiota and their products, such as bile acids (BAs), are important causal factors for the occurrence and development of CRC. Therefore, we performed 16S ribosomal RNA (16S rRNA) and liquid chromatography/mass spectrometry (LC–MS) to measure mucosal microbiota and BA composition in paired cancerous and noncancerous gut tissue samples from 33 patients with CRC at a hospital in Beijing. In cancerous tissues, we detected altered mucosal microbiota with increased levels of the genera *Bacteroides*, *Curtobacterium*, and *Campylobacter* and an increase in deoxycholic acid (DCA), which was the only BA elevated in cancerous tissues. Ex vivo coculture showed that the mucosal microbiota in cancerous tissues indeed had a stronger DCA production ability, indicating that DCA-producing bacteria are enriched in tumors. Results from the CCK8 and Transwell assays indicated that DCA enhances the overgrowth, migration, and invasion of CRC cell lines, and, through qPCR and Western blot analyses, downregulation of FXR was observed in CRC cell lines after DCA culture. We then verified the downregulation of FXR expression in cancerous tissues using our data and the TCGA database, and we found that FXR downregulation plays an important role in the development of CRC. In conclusion, differing mucosal microbiota, increased amounts of mucosal DCA, and lower FXR expression were demonstrated in cancerous tissues compared to normal tissue samples. The results of this study can be applied to the development of potential therapeutic targets for CRC prevention, such as altering mucosal microbiota, DCA, or FXR.

## 1. Introduction

The prevalence of colorectal cancer (CRC) is increasing annually in developing countries such as China. CRC is the third most prevalent neoplasm and the second leading cause of cancer death worldwide [1,2]. By 2030, it is estimated that there will be about 2.2 million new individuals with CRC, and 1.1 million people will die from CRC globally [3]. Thus, it is becoming crucial to study the causes and mechanisms of CRC [4].

An increasing number of studies have shown that gut microbiota has a vital function in the development of CRC [5,6,7]. The gut microbiota has carcinogenic or pro-carcinogenic effects through inflammation and immunological modulation, genotoxin formation, dietary metabolism, and stimulation of tumor-associated signaling mechanisms [8]. Studies have revealed that some pathogenic bacteria, such as *Porphyromonas*, *Escherichia*, *Enterococcus*, *Streptococcus*, *Peptostreptococcus*, *Bacteroides*, *Alternaria*, and *Fusobacterium*, are enriched in CRC patients [9,10,11,12,13,14,15,16,17,18,19]. Most current studies have focused on fecal microbiota due to accessibility. However, there is a considerable difference between stool and mucosa community composition [20,21,22]. The intestinal mucosa is the mediator that communicates the gut microbiota with host cells, and the mucosal microbiota exhibits an earlier reaction with the intestinal mucosa than the fecal microbiota [23]. The microbiological environments in cancerous tissues and normal gastrointestinal mucosa are different, which leads to divergences in microbial representation [14,16]. Therefore, studies on the differences in colonizing microbiota in cancerous and noncancerous tissues have profound implications for revealing the specific role of bacteria in colorectal carcinogenesis and progression.

Diet influences the incidence of CRC, especially the Western diet, characterized by a high intake of red meat and fat, which has been proven to increase the risk of CRC [24,25,26]. Studies have shown that a high-fat diet stimulates the secretion of bile acids (BAs), which are metabolized by gut microbiota [24,27]. BAs are important metabolites of the intestine and have a vital function in the development of CRC [28]. Bacteria have an essential role in BA metabolism. *Bacteroides fragilis*, *Bacteroides vulgatus*, *Clostridium perfringens*, and *Clostridium* have been associated with the metabolism of BAs [29,30]. Recent studies suggest that deoxycholic acid (DCA) and lithocholic acid may promote malignancy progression through DNA injury and the creation of reactive oxygen species [31]. Most research has focused on the structure of BAs in serum and feces. However, the expression patterns of BAs in cancerous tissues have yet to be studied thoroughly, especially studies that compare cancerous and paired noncancerous tissues simultaneously. Experiments on the action pathway of BAs in tissues compared to that in serum and feces may be more convincing in revealing the development of tumors.

Although there are several studies on bacteria, BAs, and CRC, the specific conditions in the gut mucosa and their underlying pathways are still unknown [14,16,17]. In this research, we performed 16S ribosomal RNA (16S rRNA) gene profiling of the mucosal microbiota of paired healthy colorectal mucosae and adenocarcinomas in CRC patients. By further analyzing the gut microbiota and BAs in paired samples, we demonstrated that alterations in the gut mucosal microbiota led to CRC progression by metabolizing DCA, which inhibits the expression of the farnesoid X receptor (FXR).

## 2. Methods and Materials

### 2.1. Study Design and Sample Assemblage

Individuals who were diagnosed with CRC by colonoscopy and underwent colorectal surgery at Peking University Third Hospital were enrolled in the study from November 2020 to August 2021. Pathologists histologically confirmed the disease status. The exclusion criteria were (1) patients having a history of inflammatory bowel disorder such as Crohn’s disease and ulcerative colitis, (2) other malignancies, (3) patients who used antibiotics and probiotics for several weeks prior to surgery, (4) patients with a history of intestinal resection, and (5) patients receiving neoadjuvant therapy before surgery. A total of 33 patients were included in this study. All experimentations were approved by the Medical Ethics Research Committee of our center (IRB00006761-M2020571) in accordance with the Declaration of Helsinki. All individuals signed informed consent forms before their enrollment in this research.

Polyethylene glycol electrolyte powder (IV) (Staidson (Beijing) Biopharmaceutical Co., Ltd., Beijing, China) was combined with 4 L of water for employing as a standard mechanical bowel preparation 1 day before surgery. Cancerous and noncancerous tissues were collected during surgery. Aseptic scarves were located in the operating area. To avoid potential sample contamination, air in the operating area was collected as negative contrast. A sterile saline solution was used to wash the intestinal mucosa. Aseptic surgical scissors were used to obtain three pieces of noncancerous tissues at the broken end (more than 5 cm from the tumor) and three pieces of cancerous tissue. One normal tissue sample and one tumor tissue sample were stored in a formalin tube at room temperature. The residual air and malignant and benign growths tissue samples were put in a sterilized cryovial, and liquid nitrogen was employed to cryopreserve them.

### 2.2. DNA Extraction, 16S rRNA Gene Amplification, Profiling, and Analysis

For ideal separation of bacterial DNA from malignant and healthy tissue DNA, we utilized a silica-based column following three cycles of mechanical lysis for 30 s at 30 Hz in a bead beater (TissueLyser; Qiagen) with 0.1 mm glass beads (MoBio; Qiagen, Hilden, Germany), followed by purification using a QIAamp PowerFecal Pro DNA Kit. Gel electrophoresis (1% *w*/*w* agarose in 0.5 TBE buffer) and a NanoDrop 2000 UV spectrophotometer were employed to assess the purity and amount of the separated DNA (Thermo Fisher Scientific, Waltham, MA, USA). All DNA specimens were stored at −80 °C until further use.

The hypervariable sections V3–V4 of the bacterial 16S rRNA gene were amplified using an ABI GeneAmp^®^ 9700 PCR thermocycler and the pair of primers 331F (5′–TCCTACGGGAGGCAGCAGT–3′) and 797R (5′–GGACTACCAGGGTATCTAATCCTGTT–3′) (ABI, Foster City, CA, USA). The final volume of the polymerase chain reaction (PCR) was 20 µL. It included 4 μL of 5× Fast Pfu buffer, 2 μL of 2.5 mM dNTPs, 0.8 μL of each primer (5 μM), 0.4 μL of Fast Pfu polymerase, 10 ng of template DNA, and ddH_2_O. Initial denaturation at 94 °C for 10 min was followed by 35 cycles of denaturation at 95 °C for 60 s, annealing at 68 °C for 60 s, and extension at 72 °C for 60 s, along with a single extension at 72 °C for 10 min, and finishing at 10 °C. Every sample was magnified by a factor of three. PCR product was obtained from a 2% agarose gel using the manufacturer-recommended Axy-Prep DNA Gel Extraction Kit (Axygen Biosciences, Union City, CA, USA) purification protocol and quantified with a QuantusTM Fluorometer (Promega, Madison, WI, USA).

In agreement with Majorbio Bio-Pharm Technology Co. Ltd., an equal number of refined amplicons were collected and paired-end sequenced on an Illumina MiSeq PE300 platform (Illumina, San Diego, CA, USA) (Shanghai, China). Raw sequencing reads can be found in the NCBI Sequence Read Archive (SRA) database (Accession Number: PRJNA865725).

Raw FASTQ files were demultiplexed using an in-house Perl script, before being quality-filtered by fastp version 0.19.6 [32], and combined by FLASH version 1.2.7 [33] with the following properties: (1) the 300 bp reads were cut at any location obtaining an average quality score < 20 across a 50 bp sliding window, truncated reads less than 50 bp were eliminated, and ambiguous character-containing reads were also removed; (2) only overlapping patterns larger than 10 bp were constructed according to their overlapping pattern. The greatest ratio of mismatches in the overlap zone was 0.2%. Unassimilable reads were eliminated; (3) specimens were discriminated on the basis of the barcode and primer, and the sequencing direction was altered on the basis of precise barcode matching and two mismatched nucleotides in primer matching. The sequences were then grouped into operational taxonomical units (OTUs) using UPARSE 7.1 [34] and a profile similarity threshold of 97%. The sequence with the highest frequency was chosen as the representative profile for each OTU. To limit the impact of sequencing depth on estimates of alpha and beta diversity, the count of 16S rRNA gene sequences from each specimen was reduced to around 20,000, resulting in an average Good’s coverage of 99.90%.

Specimens are susceptible to environmental and reagent contamination [35], leading to false-positive results due to low bacterial biomass. To accommodate this difficulty, during tissue gathering, the tubes were preserved open near the surgical area for the duration of the surgery. Following case-by-case confirmation of the negative controls, 16S rRNA quantification and sequencing data were employed to determine tissue-specific bacterial profiles.

### 2.3. Detection of BA

Tissue specimens were taken using a precipitation process. Chlorpropamide (C1290; Sigma-Aldrich, St. Louis, MO, USA) was employed as an internal standard for determining BA concentrations. Supernatants were evaluated for BA concentrations employing UPLC/Synapt G2-Si QTOF MS technology coupled with an ESI source, as in our previous study [36] (Waters Corp., Milford, MA, USA). The chromatographic process was performed using an Acquity BEH C18 column (100 mm 2.1 mm i.d., 1.7 m; Waters Corp.). The column was operated at 45 °C with a flow rate of 0.4 mL/min. The moveable stage comprised a combination of formic acid dissolved in water and acetonitrile at a concentration of 0.1% each. For negative mode MS recognition, gradient elution was employed. The weight range obtained was 50–850 *m*/*z*. Using standards for all BAs, the various BA metabolites revealed by liquid chromatography/mass spectrometry (LC–MS) were identified. Sigma-Aldrich was the source of CA and DCA.

### 2.4. Cell Cultures and Treatment

Malignant cell lines of the colon, such as HT-29 (HTB-38, ATCC), Caco-2 (HTB-37, ATCC), and HCT 116 (CCL-247, ATCC), were acquired from the American Type Culture Collection (ATCC, Manassas, VA, USA) and preserved in Dulbecco’s modified Eagle’s medium (DMEM, 11965; Solarbio, Beijing, China) treated with 10% fetal bovine serum (FBS, 11011-8611; Solarbio) at 37 °C in a 5% CO_2_ environment.

The cell viability was measured using a Cell Counting Kit-8 (CCK-8, K1018; Apexbio, Houston, TX, USA). HT-29, Caco-2, and HCT 116 cells were plated in 96-well plates (1 × 10^4^ cells/well, 3599; Corning, Corning, NY, USA) with DMEM, and then treated for 24, 48, 72, and 96 h with DCA at different doses (5, 10, 20, or 40 µM) and bacterial supernatants at various concentrations (5%, 10%, or 20%). Following the treatment, 10 μL of CCK-8 test solution was mixed into every well, and the cells were preserved at 37 °C for an additional 2 h. The absorbance of each well was detected at a wavelength of 450 nm.

Transwell plates were employed to quantify cell migration and invasion (3422; Corning). The bottom portion of the chamber was loaded with 600 μL of 10% FBS-containing DMEM. The top chamber filter was precoated with 50 µL of Matrigel and plated at 1 × 10^5^ cells (HCT116 and Caco-2, separately) per top chamber. The experimental cohort’s top chamber media included 200 μL of DCA solution (5 μM) produced with DMEM without FBS, whereas the control cohort contained 200 µL of DMEM without FBS. At 37 °C, cells were cultivated for 2 days. The non-migrating cells on the superficial top part of the transwell chamber were removed after being cultured and rinsed with new PBS. Underneath the membrane, invading cells were preserved in 4% paraformaldehyde and subjected to 1% crystal violet stain. An inverted microscope was employed for measuring cell count in three randomly chosen sections of stabilized cells. Every trial was repeated three times.

### 2.5. Real-Time PCR Analysis (qPCR)

Tissue specimens were kept in liquid nitrogen after being frozen. TRIzol was employed to obtain total RNA from frozen cells and tissues using the conventional phenol/chloroform extraction method. cDNA was produced using a reverse transcription kit and 2 μg of total RNA (218061; Dogesce, Beijing, China). The primer sequences for RT-PCR are summarized in Appendix A. The relative gene quantification was determined by normalization to 18S mRNA.

### 2.6. Western Blotting Analysis

Specimen and cells were homogenized in protease and phosphatase inhibitor-containing RIPA buffer. The protein extracted was electrophoretically isolated using sodium dodecyl sulfate polyacrylamide gel electrophoresis (SDS-PAGE) 218061 and then transferred to a PVDF membrane. The membranes were treated with antibodies against FXR (sc-25309; Santa, Dallas, TX, USA) and GAPDH (A10471; ABclonal, Woburn, MA, USA) overnight at 4 °C, and then preserved 1 h in a room atmosphere with HRP-conjugated anti-rabbit IgG (AS014; ABclonal). The bands were recognized employing Thermo Fisher Scientific’s SuperSignal West Dura Extended Duration Substrate utilizing the ChemiDoc XRS+ System (Bio-Rad, Berkeley, CA, USA).

### 2.7. Ex Vivo Coculture of Tissues

Cancerous and noncancerous tissues were homogenized by standing mortar and pestle in sterile PBS using aseptic techniques. Samples were inoculated (1:200) in the BHI medium in anaerobic conditions and incubated at 37 °C. After 48 h, 0.25 mL of saturated culture was transferred into 50 mL of fresh BHI medium, and sterile CA was added with a final concentration of 10 μM. The culture was incubated for 24 h and then prepared for DCA detection.

### 2.8. Statistical Analysis and Data Visualization

For 16S rRNA gene measures and alpha diversity plots [37], we used Kruskal–Wallis one-way analysis of variance (ANOVA), Dunn’s testing for pairwise comparisons, and the Bonferroni–Holm technique for *p*-value correction. Permutational multivariate analysis of variance (PERMANOVA), followed by Bonferroni–Holm *p*-value correction, was employed to determine the substantial differences between groups of 16S rRNA sequences and BAs detection shown in PCoA scatterplots. Using the LEfSe technique [38], differential abundance studies were conducted to identify the tissue-specific taxonomic characteristics that best distinguished malignant from healthy tissues. Briefly, a nonparametric factorial Kruskal–Wallis sum-rank test was initially used to identify taxa exhibiting significant differences in abundance. Subsequently, the biological relevance was explored using a series of pairwise Wilcoxon rank-sum tests among subclasses. In addition, linear discriminant analysis was employed to calculate the impact size of each differentially abundant feature [39]. A Mann–Whitney U test or Student’s *t*-test was utilized to analyze the variables of the two sample cohorts. Multiple cohort comparisons were conducted using the Kruskal–Wallis test or ANOVA. The differences in cell growth curves were uncovered using a two-way analysis of variance. The statistical significance threshold was established at *p* < 0.05. Three separate experiments provided data presented as the mean ± standard deviation (SD). GraphPad Prism, version 7.0 (GraphPad Software, San Diego, CA, USA) or SPSS, version 18 (SPSS Inc., Chicago, IL, USA) was employed for all analyses [40].

## 3. Results

### 3.1. Characterization of Enrolled Subjects

Thirty-three subjects were recruited in our study, and their characteristics are shown in Table 1. Twenty male and 13 female individuals were included, with an average age of 68.0 ± 14.6 years. Eight patients had tumors in the right colon, 18 had tumors in the left colon, and seven had tumors in the rectum. Two participants had a family history of colorectal malignancy. Both cancerous and noncancerous tissues were obtained from each patient. Cancerous tissue was defined as the cancerous group (CG), and noncancerous tissue was defined as the noncancerous group (NCG) in the subsequent analysis.

### 3.2. Mucosal Microbiota Dysbiosis Is Associated with CRC

To compare the construction of the mucosal microbiota community between the CG and NCG, we initially carried out 16S rRNA gene profiling. Regarding OTU level, alpha diversity and richness of bacterium indices showed no significant variance in the CG compared with NCG. In contrast, beta diversity analysis showed separate clusters for the CG and NCG (Figure 1A–C and Appendix A).

The relative composition of bacteria in the CG and NCG at the family and species levels is shown in Figure 1D,F. The α-diversity analysis indicated by the Shannon index was not substantially different in the two cohorts at the family and species levels (Appendix A). The β-diversity indicated by PCoA showed a considerable difference between the mucosal microbiota compositions of CG and NCG (*p* = 0.02) (Figure 1E,G). At the family level, the three highest proportions were *Lachnospiraceae*, *Coriobacteriaceae*, and *Bacteroidaceae*. At the species level, the three species with the largest proportions were *Phocaeicola vulgatus*, *Collinsella aerofaciens*, and *Escherichia fergusonii*.

Significant changes in bacteria were found by LEfSe differential analysis, which is shown in the LEfSe bar cladogram and graph (Figure 2A,B). The CG had an increased abundance of *Bacteroidaceae*, *Bacteroides*, *Curtobacterium citreum*, *Curtobacterium*, *Microbacteriaceae*, *Epsilonproteobacteria*, *Campylobacter*, *Campylobacterales*, *Campylobacteraceae*, and *Neisseria subflava*. Interestingly, *Epsilonproteobacteria* (class), *Campylobacterales* (order), *Campylobacteraceae* (family), *Campylobacter* (genus), *Microbacteriaceae* (family), *Curtobacterium* (genus), *Curtobacterium citreum* (species), *Bacteroidaceae* (family), and *Bacteroides* (genus) belonged to the same lineage, as shown in Figure 2A. The abundance of *Bacillaceae*, *Lachnospiraceae*, *Mediterraneibacter*, *Ruminococcus gnavus*, and *Pseudomonas parafulva* was significantly enriched in the NCG. Meanwhile, it was found that *Lachnospiraceae* (family), *Mediterraneibacter* (genus), and *Ruminococcus gnavus* (species) belonged to the same lineage.

### 3.3. The Content of BAs Changed Significantly in CRC

BAs, particularly secondary BAs (SBAs) composed of BAs generated by the liver during bacterial metabolism, influence the inflammation process of the intestine and carcinogenesis, according to investigations in people and animals [41]. Therefore, we determined the concentration of BAs in the tissues using LC–MS. While the concentrations of CA and CDCA decreased in the CG, DCA increased significantly in the CG as SBAs in absolute and relative proportions (Figure 3A,B). PCoA analysis revealed a substantial difference between the BA makeup of both groups (Figure 3C). VIP testing revealed considerable variations between the two cohorts in the BA structure of intestinal tissue, with DCA showing the most evident modifications (Figure 3D). Spearman’s rank correlation between all bacterial species and mucosal BAs was evaluated using the respective data acquired from the CG cohorts. A correlation heatmap showed that the levels of numerous mucosal BAs were connected with mucosal bacteria and that the degree of DCA was strongly correlated with *lipoteichoic* acid, *Roseburia inulinivorans*, and Bifidobacterium *pseudocatenulatum* (Figure 3E). Thus, Metabolites in the gut, particularly mucosal BAs, were significantly impacted by the mucosal microbiota.

### 3.4. DCA Can Be Produced by Bacteria in Cancerous Tissues

We further conducted ex vivo coculture of the samples from the CG and the NCG, and we tested the CA degradation ability and DCA production ability of the two groups. The results showed that the CG bacteria indeed had a stronger DCA production ability (Figure 4A,B). This indicated that DCA-producing bacteria were enriched in the CG. Data are the means ± SEM. Two-tailed Student’s *t*-test was applied.

### 3.5. DCA Accelerated the Overgrowth and Migration of CRC Cells

To better comprehend the influence of DCA on malignant cells, we analyzed its influence on the overgrowth and invasion of DCA. Caco-2, HCT 116, and HT-29 cells were treated with different doses of DCA (0, 5, 10, 20, and 40 μM) over a predetermined period (0 h, 24 h, 48 h, 3 days, and 4 days). The CCK-8 test demonstrated that modest doses of DCA (<10 μM) stimulated cell overgrowth 24 h after injection. Because of DCA toxicity, excessive quantities of DCA may cause widespread cell necrosis (Figure 5A–C). In addition, the transwell migrating experiment demonstrated that HCT 116 and Caco-2 cells supplemented with 5 μM DCA migrated at a greater rate, indicating that DCA increased malignant cell migration (Figure 6A,B).

### 3.6. DCA Inhibits FXR Expression in CRC Cells

FXR is a transcription factor intimately related to metabolic balance in the testicular–liver axis. FXR action is regulated by BAs, which are natural FXR ligands, and modulated by hormonal and nutritional cues [42]. The FXR target genes fibroblast growth factor 19 (Fgf19) and its short heterodimer partner (Shp) are well known [43,44]. The mRNA levels of *Fxr*, *Fgf19*, and *Shp* were quantified by qPCR. The results demonstrated that administering DCA (5 μM for 24 h) decreased the *Fxr* mRNA levels and its downstream genes (Figure 7A,B). Furthermore, the protein expression of FXR was considerably decreased in HCT 116 and Caco-2 cells supplemented with DCA (Figure 7C), suggesting that DCA suppresses the expression of FXR.

### 3.7. FXR Is Expressed at a Low Level in Cancerous Tissues

To investigate if FXR is related to clinical CRC, qPCR and Western blotting tests were performed on CRC and matching healthy tissue to evaluate the expression of FXR. The findings revealed that malignant tissues expressed FXR at lower levels than nonmalignant tissues (Figure 7D,E). Furthermore, analysis of colon malignant tumor data from the TCGA database revealed a substantial association between low FXR expression and colon cancer (Figure 7F).

## 4. Discussion

A few studies have investigated the mucosal microbiota [14,16,17]. Compared with fecal microbiota, the mucosal microbiota can better represent the microenvironment of cancerous and noncancerous tissues and is likely to have a more direct function in the pathogenesis of CRC than fecal bacteria [16,45].

According to our data, the abundances of *Bacteroidaceae*, *Bacteroides*, *Epsilonproteobacteria*, *Campylobacterales*, *Campylobacteraceae*, *Campylobacter*, *Microbacteriaceae*, *Curtobacterium*, *Curtobacterium citreum*, and *Neisseria subflava*, which could be divided into four lineages, were higher in CG than in NCG. Similar to our results, Qin et al. [46] discovered that Bacteroides were more profuse in the gut microbiota of malignant tissues. Sears et al. [47] also showed that Bacteroides comprise the bulk of CRC biofilms. Bacteroides have also been associated with the metabolism of SBAs, including debinding, oxidation, and esterification processes [29]. Recently, a randomized controlled trial in a population showed that the alteration in DCA was positively related to the proportional abundance of Bacteroides, which may be associated with its bile salt hydrolase activity [48,49]. Campylobacter was enriched in cancerous tissues, and it was found to be the passenger bacteria in CRC [50]. He et al. [51] reported that *Campylobacter jejuni* could produce cytolethal distending toxin subunit B, which has DNAse action and DNA double-strand breaks that enhance CRC development. Moreover, a new study [52] discovered that patients with a high prevalence of *Campylobacter* exhibited a Signature 3 mutation, which is related to the failure of DNA double-strand break times by homologous recombination, recommending that *Campylobacter* might enhance CRC by provoking DNA double-strand breaks.

BAs are important metabolites of the gut microbiota. Disorganized BA–microbiota crosstalk promotes CRC [53]. Gut microbiota, enriched in cancerous tissues may be involved in BA metabolism. We measured BAs in cancerous and noncancerous tissues and found that DCA was enriched in cancerous tissues, whereas CA and CDCA were exhausted. Ex vivo coculture showed that the mucosal microbiota in CG indeed had a stronger DCA production ability, indicating that DCA-producing bacteria were enriched in the CG. DCA, an SBA, is of particular concern because its hydrophobicity promotes intestinal permeability and genotoxic effects in CRC [54]. Similar to our results, DCA was substantially elevated in individuals with multiple polypoid adenomas [19]. CA and CDCA are two important primary BAs that can be converted into SBAs through deconjugation and 7-dehydroxylation of the gut microbiota. In particular, SBAs and DCA have long been postulated to be tumor promoters in the colon. Consistent with our cellular experiments, DCA has been shown to have a hyperproliferative effect and promote the migration of various CRC cell lines [55]. Recent research has revealed that DCA promotes tumor cell overgrowth, induces epithelial–mesenchymal transition, increases vasculogenic mimicry creation, and activates vascular endothelial growth factor receptor 2, leading to colorectal carcinogenesis [56]. In addition, DCA triggers CRC generated by controlling M3R and Wnt/β-catenin signaling [57].

FXR, an important BAs nuclear receptor, represents a location where environmental and genetic risk factors for CRC meet [42]. BAs may bind to FXR to control BA metabolism and cell growth [44]. In our investigation, FXR expression was substantially lower in malignant tissues compared to healthy tissues. Human and mouse intestinal cancers and the surrounding healthy mucosa have revealed that FXR expression is drastically reduced during the shift from healthy to the neoplastically altered epithelium, as suggested by earlier research [58]. Furthermore, FXR expression is negatively linked with cancer severity and poor prognosis [44,59]. It has been observed that inhibiting intestinal FXR with DCA may stimulate overgrowth and DNA injury in Lgr5^+^ cells, which can enhance the generation of colorectal cancer [44]. Furthermore, recent research has shown that FXR stimulation reduces CRC growth and invasion [60]. In our study, we found that DCA binds to FXR and inhibits FXR expression, which promotes the development of CRC.

To the best of our knowledge, this is the first investigation to compare mucosal microbiota and mucosal BAs in cancerous and paired noncancerous tissue samples. We found differences between cancerous and noncancerous mucosal microbiota in CRC patients, with cancerous tissues being enriched in bacteria that metabolize SBAs or are DNA-damaging. Conversely, depleted bacteria are mostly “beneficial” microbiota. According to our data, DCA, which promotes CRC, is increased in cancerous tissues. In addition, DCA binds to FXR and inhibits its expression; low FXR expression is associated with colorectal carcinogenesis.

Our findings identified the mucosal microbiota, DCA, and FXR as possible targets for the therapeutic approaches and prevention of CRC. In conclusion, the differences in mucosal microbiota increased DCA and lowered the FXR expression in cancerous tissues versus adjacent normal samples. These results may help to create therapeutic approaches to prevent CRC by altering the mucosal microbiota, DCA, or FXR.

## Figures and Tables

**Figure 1 microorganisms-10-02486-f001:**
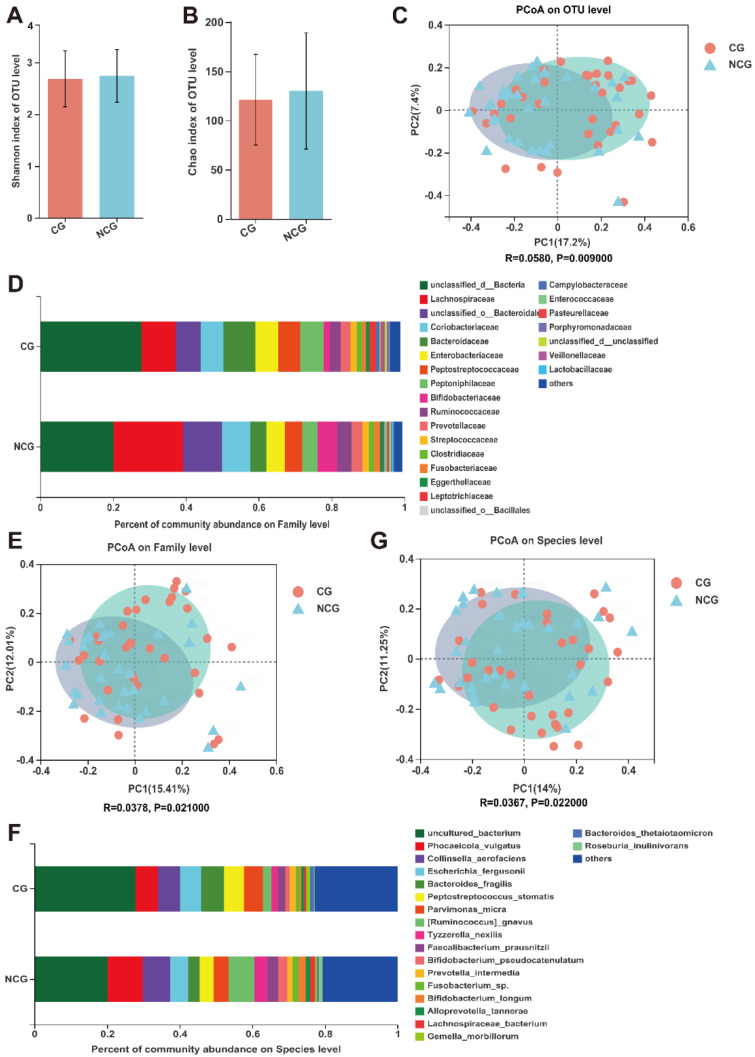
(**A**,**B**) The α−diversity analysis using Shannon index and Chao index of OTU level between cancerous group (CG) and noncancerous group (NCG). (**C**,**E**,**G**) The β−diversity analysis using PCoA at OUT level (*p* = 0.0009), family level (*p* = 0.021), and species level (*p* = 0.022) between CG and NCG. (**D**,**F**) Composition of gut microbiota at the family level and species level between CG and NCG. Greater than 1% relative abundance is shown. The smaller portions are grouped as “others”.

**Figure 2 microorganisms-10-02486-f002:**
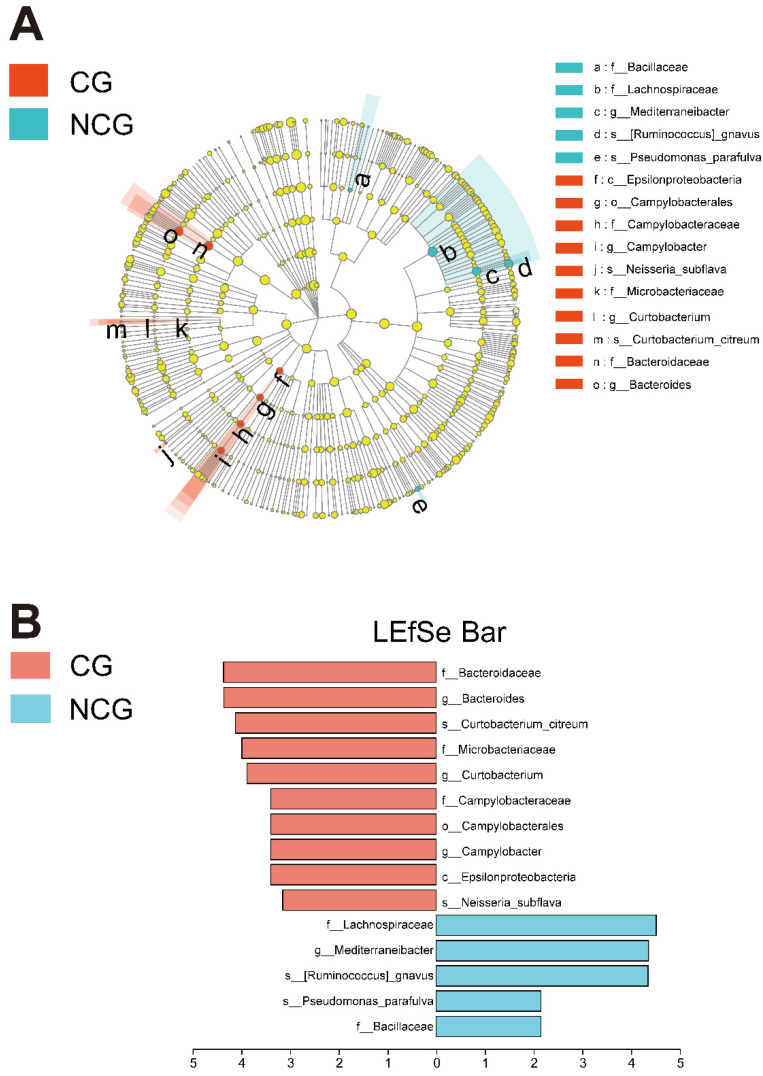
(**A**) Cladogram of differentiated bacteria between the cancerous group (CG) and noncancerous group (NCG). (**B**) Bacteria with differential abundance between the CG and NCG. Only LEfSe values over 2% are shown in the legend.

**Figure 3 microorganisms-10-02486-f003:**
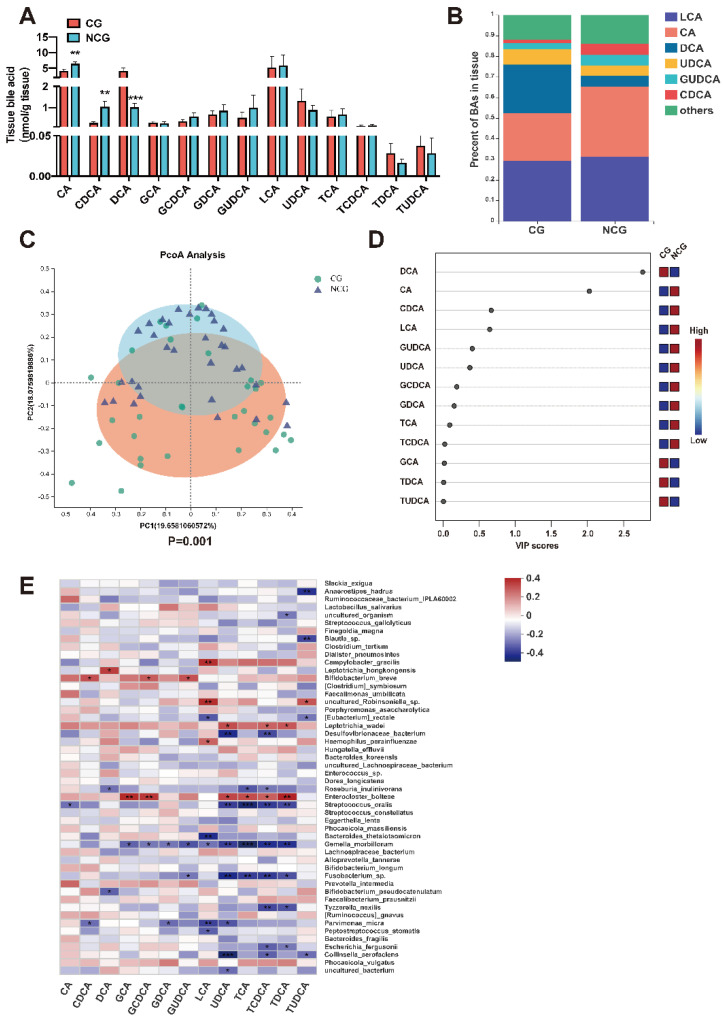
(**A**) Absolute levels of BAs in tissues between the cancerous group (CG) and noncancerous group (NCG); ** *p* < 0.01, and *** *p* < 0.001 based on one−way ANOVA with Tukey’s correction coefficient. (**B**) Proportions of tissue dominant BAs between CG and NCG. Only BAs with over 2% of total BAs are shown in the legend. (**C**) PCoA analysis of the composition of BAs between CG and NCG. (**D**) VIP analysis of BA composition between CG and NCG. (**E**) Spearman’s rank correlation analysis between mucosal BAs and gut microbiota; * *p* < 0.05, ** *p* < 0.01, and *** *p* < 0.001 based on Spearman’s rank correlation coefficient.

**Figure 4 microorganisms-10-02486-f004:**
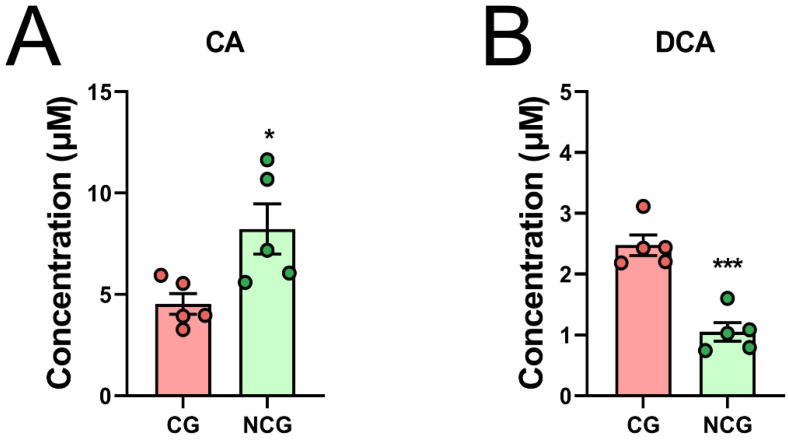
In vitro microbiota culture of cancerous and noncancerous tissues, as well as CA consuming capacity (**A**) and DCA producing capacity (**B**) in BHI medium incubated with CA (10 µM). Data are the means ± SEM; * *p* < 0.05, and *** *p* < 0.001 based on two-tailed Student’s *t*-test.

**Figure 5 microorganisms-10-02486-f005:**
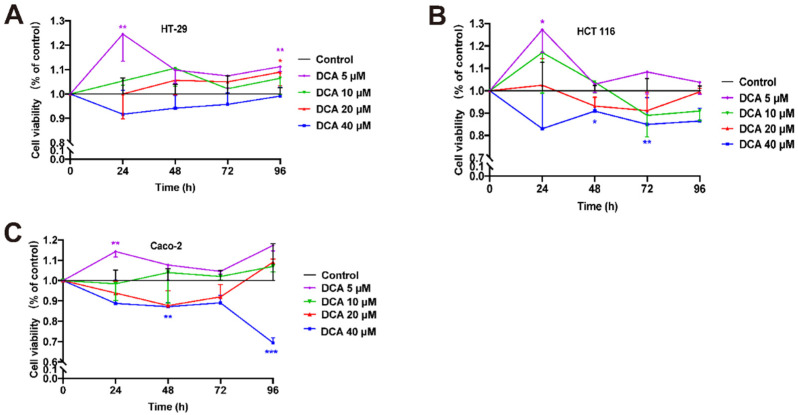
(**A**–**C**) The growth abilities of HT-29, HCT 116, and Caco-2 cells supplemented by DCA with different doses and times were detected by CCK-8; * *p* < 0.05, ** *p* < 0.01, and *** *p* < 0.001 based on a two-way analysis of variance.

**Figure 6 microorganisms-10-02486-f006:**
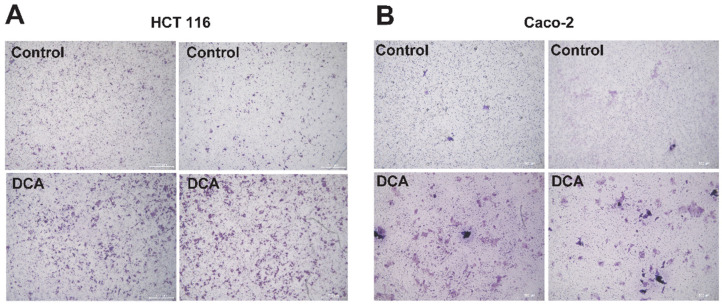
(**A**,**B**) HCT 116 and Caco-2 cells were supplemented with 5 μM of DCA for 24 h; a transwell migration assay showed that DCA accelerated migration.

**Figure 7 microorganisms-10-02486-f007:**
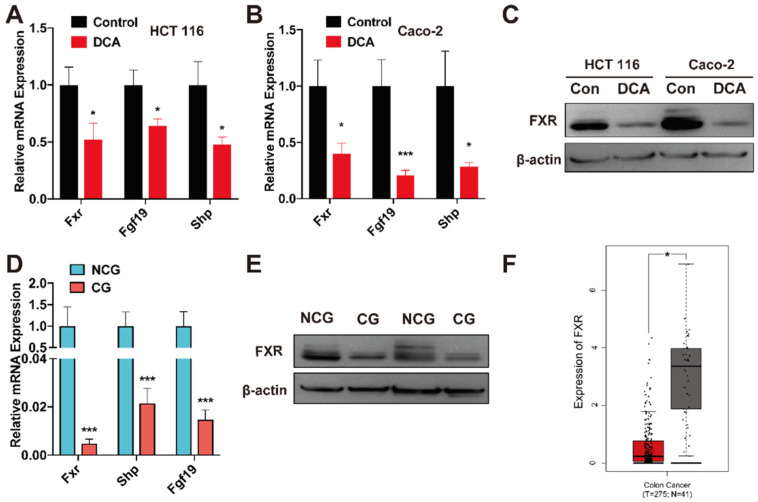
(**A**,**B**) Fxr, Fgf19, and Shp mRNA expression in HCT 116 and Caco-2 cells supplemented with or without DCA (5 μM, 24 h); * *p* < 0.05, and *** *p* < 0.001 according to one-way ANOVA with Tukey’s correction coefficient. (**C**) Western blotting bands for FXR expression in HCT 116 and Caco-2 cells supplemented with or without DCA (5 μM, 24 h). (**D**) Fxr, Fgf19, and Shp mRNA expression in tissues between the cancerous group (CG) and noncancerous group (NCG); * *p* < 0.05 and *** *p* < 0.001 according to one-way ANOVA with Tukey’s correction coefficient. (**E**) Western blotting bands for FXR expression in tissues between CG and NCG. (**F**) Data in TCGA database revealed the expression of FXR in CG and NCG tissue samples; * *p* < 0.05 and *** *p* < 0.001 based on two-tailed Student’s *t*-test.

**Table 1 microorganisms-10-02486-t001:** Clinical characteristics of the subjects.

	*N* = 33
Sex (male/female)	20/13
Age (years)	68.0 ± 14.6
BMI (kg/m^2^)	24.3 ± 4.2
Smoking history	9
Drinking history	15
CRC family history	2
Tumor site (right colon/left colon/rectum)	8/18/7
Complication	
Appendicectomy	5
Hypertension	16
DM	5

BMI, body mass index; CRC, colorectal cancer; DM, diabetes mellitus.

## Data Availability

Raw sequencing reads of 16S rRNA can be found in the NCBI Sequence Read Archive (SRA) database (accession number: PRJNA865725).

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
