# Peer review of "Promotion of Deoxycholic Acid Effect on Colonic Cancer Cell Lines In Vitro by Altering the Mucosal Microbiota"

_microorganisms, 2022, doi:10.3390/microorganisms10122486_

Round 1

Reviewer 1 Report (Previous Reviewer 1)

The abstract contains not well-written sentences. Like the second that starts with ". And microbiota". I suggest the authors double-check the text.
As per the analysis, the figures are now of much better resolution, although, for instance, Fig. 3c appears a bit weird with PC1 which puts almost all points right after 0. The same panel is used to support the sentence "Orthogonal partial least squares discrimination analysis (OPLS-DA) revealed a substantial difference between the two groups' BA makeup (Figure 3c)", but a p-value from a PERMANOVA analysis would be needed to support such statement. Also, other dimensionality reduction approaches should be considered given the strange distribution of the points in the presented plot.

Author Response

Thank you for your valuable reminder.

We have double-checked the manuscript and revised some of the statements. 

We apologized for the use of an inappropriate analytical approach here. We changed to PCoA analysis and redid the Figure 3c. We conducted the PERMANOVA analysis, and the result showed that p-value was 0.001.

Reviewer 2 Report (New Reviewer)

Ma et al found that mucosal bacteria-DCA-FXR axis plays potential pathogenic roles to CRC. Although the authors did not perform wgs analysis of the mucosal bacteria and relevant bacterial genes involved in DCA production, they did measure the CA and DCA concentrations using in vitro tissue culturing. I believe this ms will have some interest for readers of this field. I only have one minor suggestion before this ms being accepted for publication. I think supplementary  figure 3 should be part of the main figures as it conveys one import take home message and also highlighted in the abstract.

Author Response

Thank you for your kind advice.

We have put Supplementary Figure 3 into the text part.

Round 2

Reviewer 1 Report (Previous Reviewer 1)

This reviewer thank the authors for their revised version.

Author Response

Thank you!

This manuscript is a resubmission of an earlier submission. The following is a list of the peer review reports and author responses from that submission.

Round 1

Reviewer 1 Report

The manuscript by Ma Y, et al. presents a new set of CRC samples (n=330 from a hospital in Beijing analyzing the microbiome using 16S (V3-V4) and bile acids (via mass spec) from cancerous and non-cancerous gut samples.

Although the study seems overall nicely presented and organized, several things should be checked and verified.

In general, for instance, all figures appear to be of very low quality making very difficult to corroborate the statements in the text with the results in the figures.

Other points:

* Fig. 1A-B, have you considered also evaluating the richness (number of detected OTUs)? As shotgun metagenomic studies showed that CRC samples are richer than control samples.

* Lines 245-246, "Phocaeicola vulgatus, Collinsella aerofaciens, and Escherichia fergusonii" species' names should be italicized.

* The authors used LefSe but it is not cited, please double-check that also all other tools used in the work are properly referenced.

* Also Figs. 4D-E would be important to have them as separate files to verify they are free of manipulations

Author Response

The manuscript by Ma Y, et al. presents a new set of CRC samples (n=330 from a hospital in Beijing analyzing the microbiome using 16S (V3-V4) and bile acids (via mass spec) from cancerous and non-cancerous gut samples.

Although the study seems overall nicely presented and organized, several things should be checked and verified.

In general, for instance, all figures appear to be of very low quality making very difficult to corroborate the statements in the text with the results in the figures.

Thank you for your advice. We have replaced all figures with higher-quality figures in the manuscript. We also uploaded the original figures as separate files to avoid resolution loss during compression to PDF.

Other points:

* Fig. 1A-B, have you considered also evaluating the richness (number of detected OTUs)? As shotgun metagenomic studies showed that CRC samples are richer than control samples.

Thank you for your suggestion. We showed the richness (OUT level) of bacteria in the cancerous group (CG) and non-cancerous group (NCG) using the Venn figure (Supplementary Figure 2).

* Lines 245-246, "Phocaeicola vulgatus, Collinsella aerofaciens, and Escherichia fergusonii" species' names should be italicized.

Thank you for your valuable reminder. We have italicized the aforementioned species' names.

* The authors used LefSe but it is not cited, please double-check that also all other tools used in the work are properly referenced.

Thank you for your helpful advice. We have cited the LefSe [1] and alpha diversity plots [2] in the last subsection (2.7 Statistical Analysis and Data Visualization) in methods and materials.

* Also Figs. 4D-E would be important to have them as separate files to verify they are free of manipulations.

Thank you for your suggestion. We have divided Figure 4D-E into separate files (Figure 5A-B) and have changed the original Figure 5 to Figure 6 in the manuscript.

  1. Segata N, Izard J, Waldron L, Gevers D, Miropolsky L, Garrett WS, Huttenhower C: Metagenomic biomarker discovery and explanation. Genome Biol 2011, 12(6):R60.
  2. Lozupone CA, Knight R: Species divergence and the measurement of microbial diversity. FEMS Microbiol Rev 2008, 32(4):557-578.

Reviewer 2 Report

The study by Yanpeng et al. performed 16S ribosomal RNA (16S rRNA) gene profiling of the mucosal microbiota of CRC tissues and paired adjacent normal tissues. By further analyzing the gut microbiota and BAs in paired samples, they demonstrated that Bacteroidetes lead to CRC progression by metabolizing DCA, which inhibits the expression of the farnesoid X receptor (FXR). Although these are important observations, the authors failed to clearly establish a link between bacteroides, DCA and FXR in CRC, and based their conclusions on correlations rather than direct demonstrations.

Major comments

1.         Further experiments are needed to support the specificity of DCA metabolized by gut Bacteroides and its mechanism of action in CRC. To make this point, the authors should perform some rescue experiments and in vivo experiments.

2.         The authors need to focus on several specific bile acid metabolized bacteria species.

3.         Another primary concern is the modest sample size of CRC patients used to perform 16S rRNA and LC-MS studies.

4.         The English language in this manuscript would benefit from improvement for clarity and readability.

Author Response

The study by Yanpeng et al. performed 16S ribosomal RNA (16S rRNA) gene profiling of the mucosal microbiota of CRC tissues and paired adjacent normal tissues. By further analyzing the gut microbiota and BAs in paired samples, they demonstrated that Bacteroidetes lead to CRC progression by metabolizing DCA, which inhibits the expression of the farnesoid X receptor (FXR). Although these are important observations, the authors failed to clearly establish a link between bacteroides, DCA and FXR in CRC, and based their conclusions on correlations rather than direct demonstrations.

Major comments:

  1. Further experiments are needed to support the specificity of DCA metabolized by gut Bacteroides and its mechanism of action in CRC. To make this point, the authors should perform some rescue experiments andin vivo experiments.

Thank you for your valuable comment. According to the requirements, we tested the ability of various strains of Bacteroides to degrade CA and produce DCA in vitro. The results showed that these Bacteroides could not effectively degrade CA to DCA (Response figure 1A, B). However, we further conducted ex vivo co-culture of the samples from the cancerous group (CG) and the non-cancerous group (NCG) and tested the CA degradation and DCA production ability of the two groups. The results showed that the CG bacteria had a stronger DCA production ability (Response figure 1C, D), which indicated that DCA-producing bacteria were enriched in the CG.

In our sequencing results, in addition to Bacteroides, we also had increased strains, such as Curtobacterium citreum and Neisseria subflava, which will be further explored in future work.

However, our current manuscript aims to find the differences between the CRC and paired adjacent normal tissues through bacteria sequencing and bile acid metabonomics. The 16s rRNA analysis showed that various bacteria might be related to the metabolism of DCA. We have checked our manuscript carefully and found that the previous statements were easy to mislead. Thus, we have deleted the incorrect description of the “Bacteroides” in our manuscript: 1) in the last paragraph of the Introduction, “Bacteroides” was replaced with “alterations in the gut mucosal microbiota” and 2) in the fourth paragraph of the Discussion, the expression “such as Bacteroides” was deleted.

In our data, the mucosal bacteria associated with DCA metabolism were inconsistent with the bacteria in the rich set of the CRC tissue; thereby, we have only established the correlation between the bacteria and the bile acids. Please note that, as stated in our title, “Promotion of Deoxycholic Acid Effect on Colonic Cancer Cell Lines In Vitro by Alterations of the Mucosal Microbiota”, the main content of this manuscript is not to clarify the metabolic production of DCA via certain bacteria in vitro.

  1. The authors need to focus on several specific bile acid metabolized bacteria species.

Thank you for your kind advice. We have corrected the incorrect description of the “Bacteroides” in our manuscript. Moreover, this article focuses on the relationship between DCA and CRC. In the next step, we will focus on the specific mechanisms between gut microbiota and bile acid metabolism.

  1. Another primary concern is the modest sample size of CRC patients used to perform 16S rRNA and LC-MS studies.

Thank you for your comment. Because we need to obtain the patient's matching cancerous and non-cancerous tissue, and the sampling process needs to be strictly sterile, which is the reason the sample size is modest. Compared with previous studies, 33 pairs of samples have met the statistical requirements. In future studies, we will expand the sample size further.

  1. The English language in this manuscript would benefit from improvement for clarity and readability.

Thank you for your observation. We have done in-depth language editing of the manuscript, and the supporting document is attached.

Round 2

Reviewer 1 Report

Thank you for the revise version.

Author Response

Thank you for the reviewer's reply. No change has been made to the manuscript.